# Monkeypox Infection 2022: An Updated Narrative Review Focusing on the Neonatal and Pediatric Population

**DOI:** 10.3390/children9121832

**Published:** 2022-11-26

**Authors:** Francesca Gaeta, Francesco De Caro, Gianluigi Franci, Pasquale Pagliano, Pietro Vajro, Claudia Mandato

**Affiliations:** Department of Medicine Surgery and Dentistry “Scuola Medica Salernitana”, University of Salerno, 84081 Baronissi, Salerno, Italy

**Keywords:** antivirals, breastfeeding, children, monkeypox, newborn, outbreak, pregnancy, prevention, testing, vaccination

## Abstract

Monkeypox disease has been endemic in sub-Saharan Africa for decades, attracting remarkable attention only i23n 2022 through the occurrence of a multi-country outbreak. The latter has raised serious public health concerns and is considered a public health emergency by the World Health Organization. Although the disease is usually self-limiting, it can cause severe illness in individuals with compromised immune systems, in children, and/or the pregnant woman–fetus dyad. Patients generally present with fever, lymphadenopathy, and a vesicular rash suggestive of mild smallpox. Serious eye, lung and brain complications, and sepsis can occur. However, cases with subtler clinical presentations have been reported in the recent outbreak. A supportive care system is usually sufficient; otherwise, treatment options are needed in patients who are immunocompromised or with comorbidities. A replication-deficient modified and a live infectious vaccinia virus vaccine can be used both before and after exposure. Due to the persistent spread of monkeypox, it is necessary to focus on the pediatric population, pregnant women, and newborns, who represent fragile contagion groups. Here we assess and summarize the available up-to-date information, focusing on available therapeutic options, with insights into social and school management, breastfeeding, and prevention that will be useful for the scientific community and in particular neonatal and pediatric health professionals.

## 1. Introduction

Monkeypox is a sylvatic viral zoonotic (i.e., spreading between animals and humans) infection, originating from the homonymous virus (MPV), first identified by Copenhagen’s Statens Serum Institute in a cluster of monkeys imported from Singapore to Denmark in 1958. The first cases of human infection were identified in children in the year 1970 [1,2]. MPV infections were endemic in sub-Saharan Africa until its complete eradication using first-generation smallpox vaccines in 1980. In western countries, a notable epidemic of MPV infection occurred in the United States in 2003, when infected rodents from Africa, imported as pets, spread the virus to domestic prairie dogs, who later infected human subjects in the Midwest. That epidemic involved 37 confirmed cases and 10 likely cases in six states, but there were no deaths [3]. Thereafter, except in cases associated with the pet trade and travel, monkeypox has recently regained attention from health authorities, due to infections reported in the UK in May 2021 and in the US in November 2021. A few months later, (May 2022), the first outbreak was reported in continental Europe [4]. Since January 2022 and as of 17 November 2022, there have been 29,055 confirmed cases of monkeypox and 11 deaths in the United States [5]. In the same period, 25,465 confirmed cases of monkeypox and four deaths have been reported from 29 European Union countries [6,7]. Currently this global outbreak (80,221 confirmed cases and 52 deaths reported to WHO) is occurring in 110 Member States. A high proportion of these infections are from countries without previously documented monkeypox transmission and are caused by the Clade II of monkeypox, formerly known as the West African clade [7,8]. Presently, except for the South Pole, cases of monkeypox have been reported in every continent, declaring on 23 July a Public Health Emergency of International Concern (PHEIC) by World Health Organization (WHO) [9]. Despite its recent mildly declining trend [7], the current outbreak of MPV gives cause for concern also due to its long incubation period which may mask the culprit contagion and for its quite frequent subtle clinical presentation as well. Overall, epidemiological data show that the ratio M:F is 97:3, and that the population seemingly most affected (approximately 80%) is male, aged between 18 and 44 years old, mostly homosexual men having unprotected sex with other men. However, adolescents, children and babies of both sexes can be affected as well (1.2% reported cases were aged 0–17) [7,8].

Although the infection now is not yet an emergency in Western countries, the persisting outbreak of cases in recent times makes the need for comprehensive information on diagnosis, management, and therapy for all age groups seemingly reasonable. As happened for the COVID-19 pandemic, the most important world health organizations have fielded information and tools on the subject of pediatric monkeypox, including the procedures to contain contagion in pregnancy and lactation which involve the pediatric population too, in a secondary way.

In this review, we document the available evidence on the treatment and management of monkeypox infections, focusing on the neonatal and pediatric population.

## 2. Materials and Methods

Due to the recency of the multi-countries surge of the infection, this timely narrative review considered both English language scientific literature articles retrieved by Pubmed/Scopus/Google Scholar databases including the observational and intervention studies with human subjects exhibiting MPV, as well as systematic and nonsystematic reviews. Documents released as “grey literature” by the most relevant international health agencies and scientific associations made available on multiple electronic scoping searches as of 18 November 2022, were consulted as well.

## 3. Results

### 3.1. Monkeypox Virus: Generalities

Monkeypox virus is a large virus (200–250 nm), part of the group of Poxviridae (subfamily: chordopoxvirinae-genus: orthopoxvirus/species: monkeypox virus). MPV is an enveloped virus, containing a linear genome with double stranded DNA [10]. It is characterized by two phylogenetic clades: Central Africa (native to Congo Basin) and West Africa (native to Nigeria). The first clade appears more pathogenetic causing severe disease with a mortality rate of up to about 10% [11].

WHO experts reached consensus to now refer to the former Congo Basin (Central African) clade as Clade one (I) and the former West African clade as Clade two (II). Additionally, it was agreed that Clade II consists of two subclades IIa and IIb, the latter referring primarily to the group of variants largely circulating in the 2022 global outbreak [12].

In a recent study, the immunological signature of infected patients showed an early expansion of activated CD4+ and CD8+ T-cells which persisted after recovery without distinctions between HIV-positive and HIV-negative patients. Cases with mild clinical pictures showed a less perturbed immune profile early [13]. These data agree with previous studies which showed an overproduction of interleukin [IL]-2R, IL-10, and granulocyte macrophage-colony stimulating factor in patients with serious disease [14]. Cytokine response appears to be turned off in subjects who have already contracted the virus [4].

Although in the past MPV underwent few mutations, overtime, as happens for other DNA viruses, [15,16], more mutations are now increasingly being reported. These recent mutations are different from those already known previously, likely reflecting the fact that the virus is now able to survive in the population more easily [4,16].

A recent bioinformatic analysis study consistently found in the 2022 genomes two specific variation units in repeat located in intergenic regions of the viral DNA helicase and transcription factors. These factors might affect the virulence of the virus during the current outbreak [17], although the monkeypox virus continues to evolve, no mutations have affected the part of its genome that encodes a protein targeted by tecovirimat, an antiviral drug being tested for use against monkeypox in humans [18].

### 3.2. Clinical Presentation, Transmission and Prevention

The MPV final reservoir has not yet been clearly identified. Several evidences report that some rodents, originating in Africa, can transmit the virus between members of the same and different species. Additional known zoonotic reservoirs for MPV are the Gambian marsupial, tree squirrel, rope squirrel and sooty mangabey monkey, and possibly others [10]. Body fluids, droplets, scratch, bush meat preparation, lesion material wounds are some of the modes of animal trasmission [19]. Immunopathogenetic studies show that human monkeypox virus (MPV) can enter the body via the the skin or respiratory tract and starts replicating in inoculation sites. Recent data showing high viral loads on the skin and mucosa of infected patients suggest that in humans transmission most likely occurs mainly through direct contact rather than through contact with bodily fluids or respiratory transmission [20,21].

In both the above scenarios, it is hypothesized that infected Antigen-presenting cells (APC) migrate to nearby draining lymph nodes and facilitate their spread through the lymphatic system with frequent swelling of lymph nodes. After propagating through lymphoid tissue, MPV viremia can target organs such as the spleen and liver and spread further to distant organs like the skin and gonads. MPV has recently been isolated from the semen of infected individuals, underscoring the possibility of sexual transmission. MPV locates in tissues with an incubation period that lasts 7–14 days, up to a maximum of 21 days. In this step, first non-specific symptoms appear, such as fever, maxillary, cervical or inguinal lymphadenopathy, chills, headache and backache. Lymphadenopathy when present is characterized by solid consistency, sometimes painful, and tender lymphnodes. This clinical aspect could allow making differential diagnosis with smallpox as the latter does not cause lymphadenopathy [1]. At this time, patients may already be contagious. Infection of the skin and mucous membranes leads to the appearance of infectious pustules and ulcers. The latter release large amounts of virus into the saliva, possibly leading to aerosolized transmission of MPV [22,23]. MPV can also be transmitted through the placenta in vertical transmission (congenital Monkeypox), which is associated with a high risk of either perinatal loss or vertical infection, and through close contact during and after birth [24,25]. The contagion from humans to pets has been documented as well [26].

To estimate the MPV infection score, it has been used a mathematical model showing a number of reproduction (R) > 1, indicative of the epidemic potential. Quarantine can be a good strategy to prevent infection; based on the knowledge of the potential contagion, it would be useful to identify and isolate affected or potentially affected subjects up to 3 weeks [4,27]. 

After the initial contagion, and 1–2 days after the end of incubation period described above, the patient develops a rash, mainly to the face, which spreads throughout the body progressively. Typically, the lesions are initially visible within the oropharynx; afterwards they spread centrifugally to other areas of the body, including palms and soles. The rash can also be localized, and skin lesions can be from few to thousands. In a period between 2 and 4 weeks, the skin lesions change in appearance, and become macular, papular, vesicular, and pustular, most frequently all at the same time and are painful (Figure 1).

The size of the lesions is between 2 and 10 mm, remaining in the pustular phase for 5–7 days. At this point one can observe crusts that tend to desquamate in 7–14 days. In 3–4 weeks, the clinical pathological situation is resolved, and patients are no longer contagious at the fall of all the crusts [1]. In the current outbreak, affected people have however been reported with fewer lesions than in previous outbreaks in Africa, with a higher proportion of (likely more uncomfortable) hidden mucosal lesions [28].

Noticeably, in the current 2022 outbreak, the febrile prodrome may be absent and skin/mucosal lesions may be isolated to the genital and anal regions [29,30].

In addition to classical symptoms, MPV can present with rare symptoms and complications that are observed with the highest rate in immunocompromised patients. Vomiting and diarrhea can appear from the second week after infection and can cause secondary dehydration. Corneal infections are among the most complex and dangerous complications, with the possibility of vision loss. Patients affected by monkeypox can report neurological complications with rare cases of encephalitis, which need intensive supportive treatments [25,31].

Bacterial superinfection, bronchopulmonary infections, and respiratory distress can occur in immunocompromised and co-infected with influenza virus patients. Sepsis and septic shock due to bacterial over-infections can occur when the general state of the patient is impaired [32]. In view of the above complications, the MPV patients need careful monitoring.

Currently, the mortality rate is estimated between 1 and 10% in relation to the clade involved and the proper health care [33]. Quantification of environmental contamination is not well known. A study published in 2022 [34] showed that hospital environmental contamination is possible. Biological samples, collected on contaminated room surfaces examined by real-time polymerase chain reaction (RT-PCR) after a 4-day hospital stay of MPV patients, showed up to 105 viral copies/cm^2^. In view of these data, it appears important to decontaminate the hospital rooms and to protect healthcare professionals from exposure to the contagion. Currently, the following measures appear to be good practices to be implemented: (i) avoiding direct contact with wild animals or pets suspected of hosting MPV; (ii) using negative pressure rooms to isolate affected patients in hospital; (iii) killing of affected animals; (iv) avoiding contact surfaces possibly contaminated by infected persons or animals. All persons and operators who meet affected subjects or animals must wear protective equipment, such as waterproof clothing over the whole body, a double layer of gloves, and an N95 mask [4,35].

#### 3.2.1. Monkeypox in the Pediatric Population

Intrafamilial transmission has been reported in children [24] with severe and even fulminant cases in Central and West Africa [36]. Data from previous outbreaks in the African continent indicate that before the year 2000 monkeypox affected mostly young children, with a median age at presentation of 4 to 5 years. Afterwards, a trend affecting older children and younger adults (10 years of age before 2009 and 21 years in the period 2010–2019) has been observed, likely as a result of the cessation of smallpox vaccinations which left the age groups below 20 years at risk. Many of the cases involving children showed a CFR between 3.6% and 10.6% as a function of the MPV clade and the baseline nutritional status of the children [37].

According to WHO reports, the contagion of the pediatric population in industrialized regions currently remains under control, with few reported cases in the age range 0–17 years compared to the adult population. In detail, as of 16 November, of the 46,901 cases where age was available, there were 560 (1.2%) cases reported < 17-year-old, out of which 149 (0.3%) were aged 0–4. The majority of cases < 17-year-old were reported from the Region of the Americas (69%). Of the cases aged < 17 years, 23 stated exposure in a school setting. Overall, data show that the pediatric population exposed to infection can become infected as well as the adult population [7,8].

Contagion generally manifests with adult-like symptoms. Rash in children is very common as an MPV symptom, similar to the symptoms associated with common viral infections that affect children (chickenpox, herpes, allergic rashes and hand, foot, mouth diseases, molluscum contagiosum, syphilis, drug eruption) [38]. Fever, fatigue, headache, and lymphadenopathy are classic symptoms which can be difficult to diagnose in the infant population [39,40]. The interpersonal and environmental transmission pathways of children are the same as for the adult population, but the sharing of objects, clothing, or linen for children could be riskier, in light of the increased rate of unprotected contacts between children.

Although suspicion of MPV should rise when a child presents with a skin rash, a history of fever, fatigue, and lymphadenopathy, inquiring in detail the history of travel to endemic areas or contact with people from high-risk countries may represent the most crucial information at this age.

When a child tests positive for the MPV tests, all close contacts should be monitored and tested when suggestive symptoms appear. The home management of a child with MPV can be more difficult than the adult because they tend to touch his/her lesions, to spread fluids, and to contaminate the environment more easily. Moreover, children are often not independent, especially in the early years of life, and require assistance from a family member. This can increase the risk of interfamily contact and contagion. Therefore, the American Academy of Pediatrics (AAP) recommends protecting skin lesions in the child with MPV to prevent scratching or touching the eyes. In addition, it would be ideal for only a family member to have contact with the child for assistance and that no pet will ever come into contact during the contagion phase [39].

Children of at least 2 years of age should wear a mask when interacting with the caregiver, who in turn should use appropriate gloves and protective equipment to reduce infection [40]. Due to the high risk of interpersonal contagion in children’s community environments, children with MPV should avoid attending school until the end of the disease. The decision to return to school should be considered together with the local health authorities who should assess the actual risk of no infection that should be based on skin lesion healing evidence. In fact, the school environment may represent at this stage of the spread of the contagion a possible high-risk area given the close contacts occurring among children [41].

#### 3.2.2. Monkeypox in Pregnancy and in Newborns

During the current MPV outbreak, at least 21 cases were pregnant or recently pregnant; two of them were known to be hospitalized, although without need for intensive care. In 2/4 cases with a known route of transmission, MPV was transmitted via a sexual encounter in a household setting [8]. Data from endemic areas support the existence of a vertical transmission route in cases of premature births or fetal loss in mothers in the third month of pregnancy who showed the rash of MPV infection. Diagnosing suspected monkeypox in pregnant women could allow for the possible benefit of some of the available therapies for both mother and child [25]. Non-vertical but perinatally acquired MPV infection has however also been reported [42].

Due to the current scarcity of data, the best delivery mode to reduce mother-to-child transmission is still uncertain, and as with all decisions regarding caesarean section, it should continue to be a personalized decision after discussion between the expectant mother and her provider [43]. However, cesarean delivery is recommended if the mother presents with genital or anorectal lesions and/or PCR tests are not available. General anesthesia and neuraxial anesthesia considerations are necessary to be considered if oropharyngeal or cutaneous lesions near the insertion site, respectively, do exist [36,44,45,46,47].

According to the AAP [39], babies born to mothers infected with MPV during pregnancy should undergo early bathing and close observation in a special unit. They should not have direct contact with parent(s) or caregivers infected with monkeypox. Breastfeeding should be delayed during the isolation period, and breastmilk should be pumped and discarded. In high-risk neonates with positive PCR, breastfeeding may be considered provided there are no lesions on the mother’s breast [36,39,45,46].

### 3.3. Diagnosis

MPV diagnosis is based on the patient’s medical history and epidemiological data. Suspicion of MPV infection should arise in the case of travel or possible contact with wild animals from endemic areas and evidence of suggestive symptoms. When collecting the history of possible trips in endemic areas, details such as the destination and the dates of arrival and departure to assess potential incubation period and any activities carried out abroad and any exposure to people or animals potentially infected should be recorded [19]. Due to the difficulty of a differential diagnosis between smallpox and human monkeypox infection, the CDC created a specific protocol, named the “Acute, Generalized Vesicular or Pustular Rash Illness Protocol” [48]. Lymphadenopathy was used as the primary criteria to discriminate which patients are sent to second-level tests, because, as mentioned above, this was an MPV characteristic symptom [49]. However, in the present outbreak, this symptom may be absent [50].

Although history and symptoms can be indicative, it is crucial to use laboratory diagnostics, represented by the detection of viral DNA isolated from patient lesions. According to the WHO, RT-PCR specific to the MPV genome used alone, or in combination with sequencing is considered the gold standard for diagnosis in adults and children. Moreover, at all ages, the recommended specimen type for laboratory confirmation of monkeypox is skin lesion material, including swabs of the lesion surface and/or exudate, roofs from more than one lesion, or lesion crusts [42,51].

MPV infection may also be confirmed by several other newer molecular techniques, or electron microscope vision and immunohistochemistry staining for Orthopoxvirus which are reliable tests for diagnosis but are expensive or not commercially available.

Serology (IgG and IgM) with enzyme-linked immunosorbent assay (ELISA) early (after 5 and 8 days of infection for detection of IgM and IgG, respectively), is particularly important for the reconstruction of the timing of infection [1]. Antibody assay detects its presence more clearly after about two weeks of infection [1,4]. Orthopox biothreat alert © (Tetracore, Rockville, MD, USA) allows Smallpox virus antigens to be detected directly, but unfortunately, it is not able to distinguish between smallpox and monkeypox [52].

The differential diagnosis should be placed with pathologies that manifest themselves on the skin, but that have systemic involvement [53], such as smallpox, disseminated zoster, chickenpox, eczema herpeticum, scabies, and rickettsialpox. Due to MPV sexual transmission, skin manifestations of some sexually transmitted diseases (STD) such as syphilis should be considered in differential diagnosis. Furthermore, bacterial infections of wounds and skin infections can mimic monkeypox. Skin reactions to medications sometimes manifest as ulcerated lesions and monkeypox-like crusts.

Complications of monkeypox can sometimes appear as the first symptom of the disease and delay the diagnosis. A patient who typically presents with fever, characteristic lymphadenopathy, a history of exposure to risk and impaired clinical conditions (for example, pneumonia, gastroenteritis, or neurological symptoms) should arise suspicion in relation to monkeypox. Dehydration, sepsis, encephalitis, and pneumonia deserve even more attention in the pediatric population because dehydration is faster and more severe in the child than in the adult patient. As mentioned, in children lymphadenopathy may be less marked than in adults, but the history and the presence of any cases in the family should make monkeypox suspected [39].

The asymptomatic occurrence of monkeypox remains a vulnus also for the risk of vertical transmission of monkeypox, partly addressed through rigorous assessments of testing facilities in pregnant women [36,54].

### 3.4. Treatment

Several antivirals and vaccines which can be used for both smallpox and MPX are presently available. Their effectiveness and best use in patients with MPV disease are still unclear, and they are not widely accessible worldwide [55].

#### 3.4.1. Antiviral Medicines

Currently, MPV has no specific treatment; symptomatic drugs, such as non-steroidal anti-inflammatory drugs, are mainly used to control non-specific symptoms (i.e., fever and asthenia). Immunocompetent patients usually overcome the disease without treatment. Due to the similarity between monkeypox and smallpox, the CDC [56] has proposed the use of previously approved drugs for smallpox in patients who contract MPV. Treatment with antiviral drugs, according to the CDC, should be reserved for particularly compromised patients. The following three available antiviral drugs used for MPV infection have side effects and a still poorly known clinical value [57]:(a)Cidofovir (Vistide) is primarily used as a treatment for retinitis, encephalitis and oesophagitis caused by cytomegalovirus, especially in people with HIV. It is the phosphorylated active metabolite of brincidovir. In-vitro and preclinical studies showed that is effective against poxviruses [4];(b)Brincidofovir (Tembexa) is available as oral suspension/tablets, approved by FDA for smallpox disease [58]. Both drugs are inhibitors of DNA replication with a broad spectrum of activity against multiple families of double-stranded DNA viruses.(c)Tecovirimat (ST-246): is an antiviral medication which impairs the function of the VP37 envelope protein necessary for the formation of the extracellular enveloped virus required for cell-to-cell transmission; it has more specific activity against orthopoxviruses [58]. It has been approved by Food and Drug Administration (FDA), used to treat human smallpox disease but can be used against MPV. Tecovirimat is given orally (TPOXX^®^: 200 mg capsule) or as an injectable formulation [52]. Capsules should be taken within 30 min after a full meal with moderate to high fat. Per CDC guidelines, for those who cannot swallow they can be opened and mixed with liquids/soft food [59] Because it is an inducer of cytochrome P450 (CYP) 3A and CYP2B6, co-administration with this drug may lead to reduced plasma exposures of sensitive substrates of CYP3A4 or CYP2B6, reducing the effects. Because of the presence in its IV formulation of a potentially nephrotoxic substance (hydroxypropyl-β-cyclodextrin), it is advisable to dose creatinine clearance (CrCl) and liver function before starting treatment. Intravenous therapy is safe in mild/moderate renal impairment but is contraindicated in severe nephropathies (CrCl < 30 mL/min), both in adults and children [39]. Dose adjustments for oral therapy instead are not required in the case of mild, moderate, severe nephropathy or even in patients requiring hemodialysis in end-stage renal disease [60]. Although reduced fertility due to testicular toxicity was found in mouse models, no human data are available [60].

The most frequent adverse effects in oral treatment are headache and nausea, followed by abdominal pain and vomiting. The main reactions to IV tecovirimat are pain and redness at the injection site, headache, myalgia, arthritis, back pain, muscle tightness, diarrhoea, photophobia, and generalized pruritus. Recently, the largest safety study concluded was a trial on 549 mostly non hospitalized adults affected by MPV disease with/without HIV, who were prescribed oral tecovirimat under an Expanded Access Investigational New Drug protocol. The median interval was three days from initiation of tecovirimat to subjective improvement, with no difference by HIV infection status. In 3.5% of patients, adverse events were reported, one of which was nonserious. Tecovirimat is generally well tolerated, and these data support continued access to treatment with tecovirimat during the current monkeypox outbreak [61].

The effectiveness of antiviral drugs as post exposure prophylaxis in PEP monkeypox is unknown. Antiviral medications, primarily tecovirimat, can be used as PEP only in exceptional cases, for example when it is not possible to administer vaccines for allergies. To find new therapeutic strategies, studies of the efficiency of some antivirals generally used against Orthopoxvirus species are underway [62].

#### 3.4.2. Children’s Treatment

Although no study is available in the pediatric age, the use of specific drugs in paediatric patients is strongly recommended in the “severe disease”. According to the AAP, [39] the risk conditions for “severe disease” are infants, children < 8 years of age, children with eczema and other skin conditions, immunocompromise, presence of lesions on the eyes, mouth, genitals, anus (especially in adolescents), with complications from MPV [38,39,63].

Tecovirimat and Brincidofovir are authorized also for the treatment of paediatric MPV, including neonates [4,60,63,64]. Tecovirimat represents the first-line treatment and is being used under an investigational protocol. Recently, the CDC streamlined the process to obtain it. As for adults, it is available in both oral and intravenous forms [39,60]. In paediatric patients with monkeypox, treatment with tecovirimat should follow specific dosages. CDC [60] but not yet the European Medicine Agency [65] has given indications for the administration also to children of less than 13 kg body weight. (Table 1).

In patients where it is necessary to administer the drug via the nasogastric tube (NGT), this way should be preferred to the IV administration. In fact, in children it can be difficult to find effective venous access and syringe pumps. Although NGT administration is allowed under last updates in terms of therapy, to provide an alternative option in case of limited supply of IV Tecovirimat or if infusion is not feasible. At the moment, however, there are no reliable data on the enteral absorption of Tecovirimat [60].

The adverse reactions during treatment with Tecovirimat remain unknown in the pediatric age [66]. Regarding Brincidovir, pediatric recommendations [67] indicate the following dose: from 10 to 48 Kg = 4 mg/kg, once weekly for two doses; <10 Kg = 6 mg/kg once weekly for two doses.

#### 3.4.3. Treatment during Pregnancy and Breast-Feeding

To date, there are no human studies on the toxicity of Tecovirimat in pregnancy and the drug is not licensed for the treatment in this category, but it is considered safe, according to CDC protocol [60]. However, animal model studies [65] showed that there is no evidence of fetal malformations for an oral administration of 1000 mg/kg/day (23 times higher than human exposure) from the first week of gestation. Since the virus also infects by vertical way, the choice of treatment in pregnancy must be guided by the risk-benefit assessment and the dangerous complications of infection [4,64]. Considering the general virulence of smallpox, it is hypothesized that MPV may also be more virulent in pregnant women, compared to the general population, especially in the third trimester for the risk of perinatal contagion from passages in the birth canal [24].

Cidofovir is recommended only if the pregnant woman is severely ill, not as a first line of treatment. Based on existing studies of the safety/effectiveness of immunoglobulins administered, in pregnancy an antibody cocktail purified from the blood of individuals previously immunized with the smallpox vaccine can be proposed. Lastly, the limited data available on safety or efficacy of the smallpox vaccine, MVA-BN hinder evidence-based care for this high-risk group [45,46,47] (see details in next section).

Regarding breastfeeding, according to the AAP guidelines, to limit possible infection, newborns should be placed in a separate room (isolation) without contact with infected family members. Therefore, breastfeeding should be delayed during the period of isolation, and breast milk should be pumped and discarded, at least until the definition of the condition of the mother and possible maternal treatment, as mentioned above [23]. CDC guidance suggests that pregnant and breastfeeding women should be considered for treatment following CDC consultation [56].

According to the European Medicines Agency (EMA) [65], breastfeeding should be discouraged during antiviral treatment, because of the uncertainties concerning elimination in human breast milk. Animal toxicity tests demonstrate the presence of the drug in milk. Mouse model studies have shown that an oral dose of Tecovirimat <1000 mg/kg/day shows a drug-plasma ratio up to about 0.8, between 6 and 24 h after administration, up to 11 days of lactation. Therefore, a risk of toxicity to infants cannot be excluded. Otherwise, the latest CDC report about lactation [60], instead requires that the benefit risk ratio of treatment, as well as in pregnancy, is evaluated. In the presence of active lesions on the skin of the breast or chest of the mother, it may be useful to stop breastfeeding as a precaution.

Currently there are no data on adverse reactions to Tecovirimat for the pediatric population. Due to the increase in cases in the world, further effectiveness and safety studies are clearly needed. In addition to the treatment of MPV, in conditions of increased severity, it is also necessary to treat the complications. In case of lung and skin infections, therapy should be based on antibiotics at the onset of the symptoms. The management of gastroenteritis, on the other hand, must be aimed at assessing the state of hydration and subsequent integration. Sepsis, encephalitis, and eye complications should be managed at specialist hubs. In any case, data on supportive treatments for monkeypox complications are currently lacking.

### 3.5. Vaccines: Pre- and Post-Exposure Prophylaxis

#### 3.5.1. Vaccine in Adults

Due to the similarity between smallpox and monkeypox virus, the use of smallpox vaccines triggers potent neutralizing antibody responses also to MPV [68] and appears almost 85% effective in preventing MPV infection [69]. Vaccination for smallpox was not recommended from 1972 to 2022, in the USA, due to the low prevalence of the virus [52].

All presently updated smallpox vaccine available formulations offer protection against MPV infection and can be used for preexposure or postexposure (PEP) prophylaxis. However, given the limited stockpile, pre-exposure vaccination strategies are presently precluded for a mass administration and can be available only for those individuals at highest risk of MPV infection such as health care workers. Post-exposure MVA-BN clade vaccination should be reserved for those at highest risk of complications. Administration of the vaccine up to 14 days post-exposure might not prevent disease but might reduce the severity of symptoms [47,70,71].

The characteristics of the three types of smallpox vaccines that can currently be used against MPV are schematically shown in Table 2 [52].

On 22 July 2022, the European Medicines Agency (EMA) human medicines committee (CHMP) recommended extending the indication of Imvanex to include protecting adults from monkeypox disease.

As recently experienced during the SARS-CoV-2 pandemic, there is still diffidence in the general population towards vaccines. Therefore, it is essential that global health agencies encourage the production and administration of vaccines that have been available for many years. Many studies have shown that the use of MVA-BN is safe in immunocompromised populations such as those with AIDS [72,73].

The animal studies also demonstrated the safety of the vaccine in pregnancy, with no evidence of malformations.

#### 3.5.2. Vaccines in the Pediatric Population

In a randomized controlled study of children aged between 1 and 17 years, the updated version of MVA was safely used as a carrier for Ebola protein [74]. In view of this evidence, the use of MVA in this age group would be desirable. However, as for adults, there is currently no monkeypox vaccine sufficiently available for a pediatric universal prophylactic vaccination.

Data on post exposure prophylaxis (PEP) to prevent monkeypox in children are limited and there are no vaccines or other products currently authorized for PEP in paediatrics specifically designed and approved against MPV. The choice to perform a PEP must be guided by the level of risk and the state of the patient. The MVA JYNNEOS vaccine may be recommended for children < 18 years of age for PEP under expanded use authorization issued by FDA. According to the AAP, clinicians should discuss the use of vaccine in a child as post-exposure prophylaxis with the local health department (Extended Use Authorization issued by the Food and Drug Administration) within 4–14 days from exposure [39,75]. Children and adolescents with exposure to people with suspected or confirmed MPV infections may be eligible for PEP with vaccination [71,76].

#### 3.5.3. Vaccines during Pregnancy

There are no clinical efficacy data regarding women who are pregnant or breastfeeding. However, the CDC stated on 1 September 2022, pre- or post-exposure prophylaxis could be offered to women who are pregnant or breastfeeding with the live, non-replicating viral vaccine which cannot spread to an unborn child or a breastfed newborn, as also endorsed by the UK authorities [47,77].

### 3.6. Post-Exposure Measures

Regarding Immune globulin or antiviral drugs, there are not many PEP studies on children < 16 years, but the emerging scenario of MPV requires using VIGIV, as recommended by CDC [62]. In extreme cases, Tecovirimat may be considered for use as monkeypox PEP as in documented allergy to vaccine or past reactions to immunoglobulins

In infants born to a suspected or confirmed mother with MPV infections, there are not much data on post exposure prophylaxis. A specific optimal strategy has not yet been defined but Vaccinia Immune Globulin Intravenous (Human) (VIGIV)—CNJ-016- [77], should be considered, since it has been approved for treatment of smallpox vaccine complications. Because children under 6 months of age might present a decreased response to PEP—vaccination, vaccinia immunoglobulin may be offered as an alternative PEP model [62]. VIGIV is administered at a dose of 6000 U/kg, indicated in post-vaccination attenuated live virus activation reactions, as soon as the symptoms appear. It is necessary to increase the dose (9000 U/kg/24,000 U/kg) if the patient does not respond [77].

### 3.7. Population, Ethics and Risk of Discrimination

The media and international health/scientific societies attention to the recent spread of monkeypox has opened the discussion on some ethical issues in the management of emerging infectious diseases.

In the 1980s, the new emerging HIV virus created a stigma in the black and homosexual population, as well as in the population of the African continent, from which the origin of the virus derives. At the end of 2019, the preliminary considerations on coronavirus focused instead on the population of Chinese origin and eating habits. In 2022 with the spread of monkeypox, to understand the spread of the virus in the world, international health and scientific societies have tried to analyze the contagion according to different routes, including sexual transmission, known to be a vehicle of infection. This has allowed create reports to be created that also take into account men to men or women to women relationships and not only in the heterosexual population [8]. The reported sexual orientation does not necessarily reflect who the case has had recent sexual history with nor does it imply sexual activity. The need to stratify populations according to sexual orientation hides the risk of spreading wrong messages on monkeypox. It is obvious that the mode of contagion of an infectious disease should no way affect us in the management of the patient and that no patient is more or less deserving of treatment based on how and where he contracted an infection [78].

Community education is extremely important to prevent the spread of new infectious diseases, especially when validated vaccines are available. Furthermore, the safety of MVA-BN is also validated in HIV patients; therefore, the access to vaccines must be encouraged also in these categories [24]. An editorial published in August 2022 focuses on the language to be chosen when talking about monkeypox. The nomenclature “monkeypox”, is currently being revised by the WHO which recognizes the risk of hidden discrimination [78,79].

Another risk of discrimination is that, despite the accessibility of vaccines. The largest stockpile is owned by or contracted to the United States, as they contributed to develop the vaccine as a defense strategy after 9/11 for fear that MPV could be used as a bioweapon. There are no doses purchased or ordered for African countries to date. In fact, as of 12 September 2022, the MVA-BN (Jynneos, IMVANEX, MVAMUNE) vaccine was available in Australia, Brazil, Canada, Caribbean, Cyprus, Democratic Republic of Congo, Denmark, Europe, France, Germany, Iceland, Indonesia, Israel, Italy, Mexico, Nigeria, Norway, Portugal, Scotland, Spain, and the U.K. [80].

Last, but not least, also the PCR testing confirmatory diagnosis of MPV infection itself is accessible prevalently to the more-resourced countries. The actual death rate is therefore likely higher than current estimates for underdiagnosis during this outbreak. One should fear that it could still rise, especially if the virus spreads more extensively among people at high risk of severe disease, such as children, older people, and those with a severely compromised immune system.

## 4. Conclusions

MPV is an emerging zoonosis in the Western countries. Generally, severity has been low, with few reported hospitalizations and deaths. However, while the availability of a vaccine, the generally mild presentation of the disease, and its fairly low transmissibility argue for an epidemic situation still under control, the broad host range of monkeypox virus raises concerns about the possible establishment of new reservoirs [81]. Moreover, although MPXV is not as deadly or contagious as the variola virus that causes smallpox, it poses a threat because it could evolve into a more potent human pathogen [82].

Therefore, the need to be prepared to manage the clinic and therapy of a possible increase in the outbreaks is a priority. In all cases, physicians should not only maintain a high level of clinical suspicion for MPV disease, but also follow the available protocol for diagnosing, reporting, and isolating cases, and allaying public fears and misconceptions.

The most recent Technical Report of CDC (October 2022) observed that the outbreak is slowing as the availability of vaccines has increased and people have become more aware of how to avoid infection. Still, it is their opinion that monkeypox is unlikely to be eliminated from the U.S. in the near future [83].

Undertreatment and underdiagnosis availabilities in under-resourced countries remain a relevant issue needing urgent attention. Particularly, due to the low prevalence in the paediatric population, it may happen that a pediatrician may fail to recognize and manage patients with potential MPV infection. Knowing the clinical signs, differential diagnosis with other common pediatric exanthemas, and treatment of children with MPV can help in the infection containment, and its early treatment which is needed especially in the most severe forms.

As with other physicians, pediatricians and obstetricians need to be aware that the clinical manifestations of MPV are wide-ranging. Due to the variety of the first and more apparent signs, all clinical departments could become the first point of contact for MPV cases in the general hospitals as well as in the children’s hospitals [84].

The choice of therapeutic strategies, including the treatment of complications, and the management of side effects must be within the reach of all personnel working in the first intervention, especially with the pediatric population. In conjunction with these challenges, stigma and discrimination also negatively affect case detection and may put some communities at an avoidable increased risk of worse outcomes and continuing transmission.

## Figures and Tables

**Figure 1 children-09-01832-f001:**
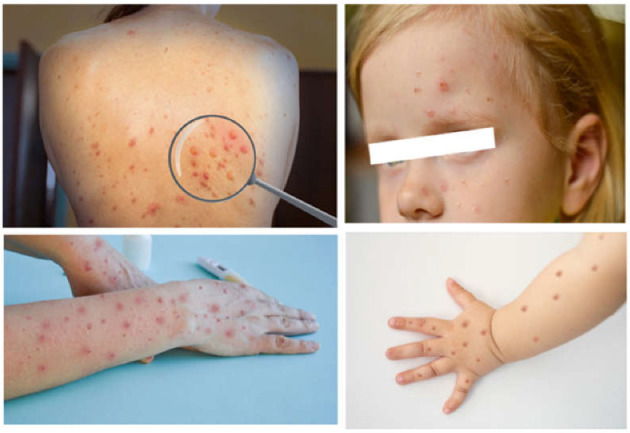
Diverse phases of the evolution of the monkeypox rash in adults (**left**) and in children (**right**) Photo credit: Shutterstock.

**Table 1 children-09-01832-t001:** Treatment with Tecovirimat in children.

Oral **	Intravenous
- from 13 to 25 kg: 200 mg q12 h for 14 days; - from 25 to 40 kg: 400 mg q12 h for 14 days; - >40 kg: 600 mg q12 h for 14 days. The latest update published by the CDC (version 6.2 24 October 2022) on the use of Tecovirimat in children [60] provides additional indications of its use in children from birth: - <3 kg: 33,3 mg q12 h - <6 kg: 50 mg q12 h; - from 6 kg to 13 kg: 100 mg q12 h. - from 13 kg, the dosages are in keeping with those indicated by the EMA.	<35 kg: 6 mg/kg by IV infusion in <6 h. Volume of infusion depends on the weight. In children < 2 years of age, - renal function should be monitored during treatment given the drug potential accumulation due to renal immaturity, - there are no data on the risks derived from hydroxypropyl-β-cyclodextrin, an ingredient in IV Tecovirimat. For children <3 kg the drug is allowed according to the CDC report on treatment of 24 October 2022, but dose adjustments may be necessary in infants based on the general condition and weight [60].

** According to CDC, treatment duration should be from 14 days to 90 days depending on the progression of the disease and clinical condition of the patient. In the pediatric population, the drug can be dissolved in breast milk, as well as in fresh foods and liquids. The failure of therapy in children may depend on the inability to take food prior to taking the drug, especially in children with impaired swallowing. Still, for doses below 200 mg the risk of dosing errors in the opening of the capsule is greatly increased.

**Table 2 children-09-01832-t002:** Synopsis of the Smallpox vaccines that can currently be used against MPV.

ACAM2000	MVA-BN **	LC16m8
(Live vaccinia virus) Second-generation vaccine	(Attenuated, non-replicating vaccinia virus)	(Modified vaccinia virus)
Administration by multiple percutaneous puncture device of the skin surface. Single dose with lesion at the site of inoculation. Not recommended in A. Individuals with - immunocompromise - atopic dermatitis B. Pregnant women. Possible adverse cardiac reactions.	Administration in two subcutaneous doses, 4 weeks apart, no lesions at the inoculum site. Safe for immunocompromised patients because the virus does not replicate. If not large available, a single dose may be administered.	Single dose administration. Licensed in Japan Safer then ACAM2000, given the lower replication capacity.

** The two-dose Modified vaccinia Ankara—Bavarian Nordic [MVA-BN (JYNNEOS^®^, USA, IMVANEX^®^, Europe; IMVAMUNE^®^, Canada)] was approved by the U.S. Food and Drug Administration (FDA) on 24 September 2019, and by the European Medicines Agency EMEA/H/C/002596 in 2013 and is now indicated for preventing smallpox and monkeypox diseases in adults 18 years of age and older determined to be at high risk for infection. In Canada and in the USA, it is licensed for both smallpox and monkeypox in adults, but not in children.

## Data Availability

Not applicable.

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
