# Peer review of "Monkeypox Infection 2022: An Updated Narrative Review Focusing on the Neonatal and Pediatric Population"

_children, 2022, doi:10.3390/children9121832_

Round 1

Reviewer 1 Report

In general the manuscript is well-written. The review is important in the current outbreak context. However, some points should be revised: 

- should consider changing the title of point 3.2 (line 110) to clinical presentation, transmission and prevention;

- please confirme that western blot analysis (line 246) is used for diagnosis. The articles referring to this technique do not present a coherent bibliography (https://pubmed.ncbi.nlm.nih.gov/35760647/; https://www.acpjournals.org/doi/full/10.7326/M22-1581). WHO does not refer this technique in their interim guidance (see https://www.who.int/publications/i/item/WHO-MPX-laboratory-2022.1);

- should be mentioned which are the best samples to colleted for monkeypox diagnosis in newborns with symptoms (see reference https://www.nejm.org/doi/full/10.1056/NEJMc2210828) and no symptoms.

Author Response

We thank the Reviewers for their very valuable comments.

PLEASE SEE THE ATTACHMENT WITH OUR RESPONSE. 

Reviewer 2 Report

This work is comprehensive and of great significance. The topic of your manuscript is very intriguing and timely. The authors provide more details about MPV. The review is well written, but some points should be stated and sections need to be rephrased. From my point of view, the following comments and suggestions might improve the overall knowledge and the entire manuscript

Firstly, the English language grammar and style need proofreading. Fig 1 is informative, have you received permission for it? 

Please cite articles and studies to show the prevalence of MPV among patients according to gender and age groups in the introduction, or section 3.1. and report their findings in numbers.  

Ref 13, the article is accepted for publication, cite the published article directly.   

Reorganize table 2 to make it easier to read. 

In the end, please take some time to reorganize and reformat the flow of the review, check previous reviews regarding MPV and follow their track. 

Author Response

  • We thank the Reviewers for their very valuable comments.
  • Please see the attachment.
